# Use of extracellular vesicle microRNA profiles in patients with acute myeloid leukemia for the identification of novel biomarkers

**Ka-Won Kang**[1☯], **Jeong-An Gim**[2☯], **Sunghoi Hong**[3], **Hyun Koo Kim**[4], **Yeonho Choi**[5], **Ji-ho Park**[6], **Yong Park**[1]*

**1** Department of Internal Medicine, Division of Hematology-Oncology, Korea University College of Medicine, Seoul, South Korea, **2** Department of Medical Science, Soonchunhyang University, Asan-si, South Korea, **3** School of Biosystem and Biomedical Science, Korea University, Seoul, South Korea, **4** Department of Thoracic and Cardiovascular Surgery, Korea University College of Medicine, Seoul, South Korea, **5** Department of Bio-convergence Engineering, Korea University, Seoul, South Korea, **6** Department of Bio and Brain Bioengineering, Korea Advanced Institute of Science and Technology (KAIST), Daejeon, South Korea

☯ These authors contributed equally to this work.
* paark76@hanmail.net

**Data Availability Statement:** All relevant data are within the paper and its Supporting Information files. The datasets generated and/or analyzed

## Abstract

### Objectives

This study aimed to establish clinically significant microRNA (miRNA) sets using extracellular vesicles (EVs) from bone marrow (BM) aspirates of patients with acute myelogenous leukemia (AML), and to identify the genes that interact with these EV-derived miRNAs in AML.

### Materials and methods

BM aspirates were collected from 32 patients with AML at the time of AML diagnosis. EVs were isolated using size-exclusion chromatography. A total of 965 EV-derived miRNAs were identified in all the samples.

### Results

We analyzed the expression levels of these EV-derived miRNAs of the favorable (n = 10) and non-favorable (n = 22) risk groups; we identified 32 differentially expressed EV-derived miRNAs in the non-favorable risk group. The correlation of these miRNAs with risk stratification and patient survival was analyzed using the information of patients with AML from The Cancer Genome Atlas (TCGA) database. Of the miRNAs with downregulated expression in the non-favorable risk group, hsa-miR-181b and hsa-miR-143 were correlated with non-favorable risk and short overall survival. Regarding the miRNAs with upregulated expression in the non-favorable risk group, hsa-miR-188 and hsa-miR-501 were correlated with non-favorable risk and could predict poor survival. Through EV-derived miRNAs–mRNA network analysis using TCGA database, we identified 21 mRNAs that could be potential poor prognosis biomarkers.

during the current study are available in the Sequence Read Archive of the NCBI repository (accession number PRJNA923832).

**Funding:** This research was supported by a grant from the Seoul R&BD Program through the Seoul Business Agency (SBA) funded by the Seoul Metropolitan Government (grant number: BT210040), and the National Research Foundation of Korea (NRF) grant funded by the Korean Government (MSIT) (grant number: NRF-2022R1G1A1006030). This research was also supported by a grant from Korea University (grant number: K2110281). The funders had no role in study design, data collection and analysis, decision to publish, or preparation of the manuscript.

**Competing interests:** The authors have declared that no competing interests exist.

**Abbreviations:** AML, Acute myeloid leukemia; BM, bone marrow; EV, Extracellular vesicle; miRNA, MicroRNA; mRNAs, messenger RNAs; NTA, nanoparticle tracking analysis; SEC, size-exclusion chromatography; TEM, transmission electron microscopy; UV, ultraviolet.

## Conclusions

Overall, our findings revealed that EV-derived miRNAs can serve as biomarkers for risk stratification and prognosis in AML. In addition, these EV-derived miRNA-based bioinformatic analyses could help efficiently identify mRNAs with biomarker potential, similar to the previous cell-based approach.

## Introduction

MicroRNAs (miRNAs) are small non-coding RNA molecules composed of approximately 22 nucleotides, which target messenger RNAs (mRNAs) for cleavage or translational repression, and thus play a role in RNA silencing and regulation of post-transcriptional gene expression [1, 2]. Based on these regulatory functions, miRNAs act as oncogenes or tumor suppressor genes and are also involved in the initiation and progression of human malignancy [3–5]. Extracellular vesicle (EV)-derived miRNAs can be detected in many types of body fluids, such as blood, urine, saliva, and cerebrospinal fluid [6–8]. Given that miRNAs can be selectively packed into EVs according to the status of parental cells and that EV-encapsulated miRNAs are highly protected from degradation, these miRNAs may be better biomarkers than non-EV-encapsulated miRNAs [9–11]. Therefore, EV-derived miRNAs are promising biomarkers for cancer diagnosis and treatment. This is a highly relevant emerging field of research [12, 13].

Acute myeloid leukemia (AML) is a malignant, clonal disorder that originates from hematopoietic progenitor cells. It is characterized by the accumulation of somatically acquired genetic alterations [14–16]. Various gene mutations, dysregulated expression of genes and non-coding RNAs, and epigenetic changes are related to its pathogenesis, subtype classification, prognosis prediction, and treatment targets; aberrant expression of miRNAs is one such factor [14–16]. In AML, miRNA expression profiling has been shown to aid in distinguishing between morphologically different AML and acute lymphoblastic leukemia [17]; help stratify morphological sub-classes of AML [18], cytogenetic subtypes, or molecular aberrations [19–22]; and is also an efficient prognostic marker [23, 24]. Furthermore, differentially expressed genes correlated with differentially expressed miRNAs may be relevant to the biological pathways or survival of patients with AML [25, 26]; however, very few studies have focused on EV-derived miRNAs in AML.

Several studies have suggested that EV-derived miRNAs may be involved in the pathogenesis of AML and may be useful biomarkers for the early detection of AML recurrence or prognosis [27–31]. For example, previous studies reported that the expression levels of miR-26a-5p and miR-101-3p were significantly increased in EVs of AML-derived mesenchymal stromal cells compared to those in cells from healthy controls. Moreover, these EV-derived miRNAs with upregulated expression could predict the altered gene expression associated with leukemogenesis (EZH2 and GSK3β) in CD34+ cells from patients with AML [27]. EV-derived miRNAs from AML cells may also be involved in compromised hematopoiesis in AML [28, 29]. In addition, EV-derived miRNAs from AML cells may be useful as biomarkers for prospective tracking and early detection of AML recurrence or poor prognosis predictors in patients with AML [30, 31]. However, most of these studies were conducted in laboratories (cell line-based or animal studies) and used EV isolation methods, such as sequential ultracentrifugation or precipitation methods, with a relatively low purity [32, 33]. Moreover, most previous studies focused on miRNAs and analyzed their potential as biomarkers; there is paucity of data on the mRNAs associated with these miRNAs.

Therefore, in this study, we isolated EVs from the bone marrow (BM) aspirate of patients with AML using size-exclusion chromatography (SEC), a high-purity method, and then analyzed EV-derived miRNAs. Using bioinformatics analysis, we investigated the role of miRNAs (which have mRNA regulatory functions) as a biomarker for poor prognosis in patients with AML.

## Materials and methods

### Patient and sample collection

BM aspirates were collected from 32 patients with AML at the time of diagnosis. Briefly, a bone marrow aspirate of 5–10 mL was collected at the time of the bone marrow examination for the patient's diagnostic purposes. Immediately after collection, the aspirate was placed in a serum-separating tube or an ethylenediaminetetraacetic acid tube and stored at 4˚C. Within one hour, the collected sample was centrifuged at 3000 rpm for 10 minutes, and 0.5-ml portions were aliquoted into 1.5-ml microtubes. All procedures were performed at 4˚C. Subsequently, the samples were stored in a deep-freezer until the experiment, with a single thawing performed just before the experiment. The samples used in this study were collected between May 2008 and February 2017, and experiments using these samples were conducted between 2018 and 2020.

The baseline patient characteristics are presented in **Table 1**. Risk stratification of patients with AML was performed using the 2017 European LeukemiaNet recommendations, which are based on cytogenetic and molecular abnormalities [34]. The risk group was divided into the favorable and non-favorable groups, and the non-favorable group comprised patients with intermediate and adverse risks. Risk stratification in AML plays a critical role in determining treatment strategies and predicting outcomes [35–37]. It is well known that favorable-risk

**Table 1. Baseline characteristics of patients with acute myeloid leukemia.**

| Base characteristics | Total patients (*n* = 32) |
|---|---|
| Median age, years (range) | 53.5 (17.0–74.0) |
| Sex (male) (%) | 17 (53.1) |
| Induction chemotherapy, *n* (%) | |
| Idarubicin-based | 20 (62.5) |
| Daunorubicin-based | 12 (37.5) |
| Risk stratification by cytogenetic and molecular abnormalities, *n* (%) | |
| Favorable | 10 (31.2) |
| Intermediate | 18 (56.3) |
| Adverse | 4 (12.5) |
| Transplantation done, *n* (%) | 17 (53.1) |
| Autologous bone marrow transplant | 4 (23.5) |
| Allogeneic bone marrow transplant | 13 (76.5) |
| Overall survival, months (range) | 18.5 (1.8–141.5) |

The experimental protocols were approved by the internal review board (IRB) of Korea University Anam Hospital (IRB No. 2015AN0267). All human sample collections and procedures were performed in accordance with the ethical standards of the IRB of Korea University Anam Hospital and the 1964 Helsinki Declaration and its later amendments or comparable ethical standards. The sex, age, and medical information of each patient were obtained, but personal information was not collected. All information was anonymized to ensure that individual participants could not be identified. Written informed consent was obtained from all participants.

AML has low relapse and high survival rates owing to induction and consolidation chemotherapy alone. On the other hand, in intermediate- or adverse-risk AML, the consideration of stem cell transplantation is essential in the treatment decision. Therefore, in this study, we categorized the groups as favorable and non-favorable, according to the different treatment approaches considered in clinical practice.

## Isolation of EVs

In the case of blood, especially plasma, interferences may be caused by lipophilic or hydrophilic proteins in the blood. To address this issue, we used commercial SEC (EXo-I, Exopert, South Korea) with two-layer column beads consisting of CL-6B sepharose and sephacryl s-200 High Resolution to isolate EVs, which allows the exclusion of lipoproteins and soluble proteins (molecular weight, 35–75 kDa) from plasma, while maintaining the integrity of EV markers [38]. This method, which has been widely used in various studies, effectively separates EVs [39–42] and is briefly summarized as follows. A total of 0.5 mL of BM aspirate serum or plasma from each patient with AML was centrifuged at $10,000 \times g$ at 4˚C for 30 min to eliminate impurities, and the resulting supernatant was loaded onto the column. Phosphate-buffered saline (PBS, w/o calcium, magnesium chloride) was used as the elution buffer, and 0.5 mL of the column eluent was collected for every fraction. The eluted fractions (11 and 12; 0.5 mL each) were concentrated using an Amicon® Ultra 100 kDa filter with a molecular weight cut-off of 100 kDa (Merck Millipore, Temecula, CA, USA), according to the manufacturer's instructions. These were used subsequently.

## Sizing and evaluation of EVs

Sizing and evaluation of EVs were performed using nanoparticle tracking analysis (NTA) and transmission electron microscopy (TEM). TEM was performed using a Tecnai G2 F20 (FEI, OR, USA) instrument at 200 kV. TEM specimens were prepared as follows [43]. The carbon grid (300 mesh) was treated with ultraviolet (UV) light for 15 min. The grid was then immersed in 15 μL of exosome samples on Parafilm. After 10 min, the grid was transferred to a PBS droplet thrice for washing. Next, the grid was fixed with 2.5% glutaraldehyde in PBS. After 10 min, the grid was washed thrice with deionized water droplets. The remaining solution was removed by gently touching with a paper wipe and drying thoroughly. For NTA, a Nanosight NS300 (Malvern Panalytical Ltd., Malvern, UK) was used. The dynamic motion of exosomes was recorded and analyzed using NTA 3.4 software (Malvern Panalytical Ltd., Malvern, UK).

## Protein extraction and western blotting

The protein concentration was determined using a BCA protein assay kit (Pierce, Rockford, IL, USA). The protein concentration as determined using the BCA assay was 8877.141 μg/mL in serum and 8965.662 μg/mL in plasma. In total, 20 μg of each protein sample were separated using 10% SDS-PAGE. The resolved proteins were transferred onto a 0.2-um PVDF membrane (Bio-Rad, Hercules, CA, USA). After blocking with 5% skim milk (w/v) in 0.1% TBST for 2 h, the membranes were probed overnight at 4˚C with 1:1000 dilutions of rabbit anti-CD63 polyclonal antibody and rabbit anti-CD81 polyclonal antibodies (all from Bioss Antibodies, MA, USA). Peroxidase-conjugated anti-rabbit antibody (1:1000; Cell Signaling Technology, Beverly, MA, USA) was used as the secondary antibody. The antibody–antigen reactions were visualized using Western ECL substrate (Bio-Rad). Images were acquired using an Amersham ImageQuant 800 Western blot imaging system (Cytiva, Little Chalfont, UK), with exposure times of 2 min for CD63 and 11 min for CD81.

## RNA isolation and RNA expression profiling

Total RNA was isolated from each sample using the miRNeasy Serum/Plasma Kit (Qiagen, Hilden, Germany). Each sample (30 μL) was mixed with QIAzol lysis buffer (1 mL) and the mixtures were processed according to the manufacturer's guidelines. The quality and quantity of the RNA obtained were verified using an Agilent 2100 Bioanalyzer with RNA Pico and small RNA kits (Agilent Technologies, Santa Clara, CA, USA). Libraries were prepared using the SMARTer smRNA for Illumina kit (Takara Bio, Shiga, Japan) according to the manufacturer's instructions, and sequencing was performed using Illumina® HiSeq 2500 (Illumina, San Diego, CA, USA) to generate 51 base single-end (1 × 50 base pairs) reads for small RNA sequencing. FASTQ files were used for primary data analysis. NGS data produced as FASTQ files were obtained from Macrogen (Seoul, Korea).

## Bioinformatic analysis of miRNA sequencing data

Following sequence alignment, known and novel small RNAs were retrieved using the miR-Deep2 software algorithm. For sequence alignment, we used the human reference genome release hg19 from the UCSC Genome Browser. The reads were aligned to the precursor, and mature human miRNAs were obtained from miRBase. Transcript abundance was measured in fragments per kilobase of exon per million fragments mapped (FPKM). The reads for each miRNA were normalized to log2 (FPKM+1). Data processing and visualization were conducted using R 4.0.3 (http://www.r-project.org). Differentially expressed miRNAs were selected from the total miRNAs between the two groups under one objective condition (favorable vs. non-favorable) as the thresholds of fold changes and p-values. We used "*t*-test," a default R function, to obtain differences as fold changes and p-values between two groups using an in-house source code. We also generated a heatmap using the pheatmap R package, measured the expression levels for each miRNA, and performed hierarchical clustering analysis.

## The cancer genome atlas (TCGA) dataset application

The mRNAs and miRNA expression data were downloaded from TCGA database (https://portal.gdc.cancer.gov/). Analyses were performed using the R package TCGAbiolinks and the GDCquery function with the following parameters:

- For mRNA data: project, TCGA-LAML; data category, transcriptome profiling; data type, gene expression quantification; and workflow type, HTSeq-FPKM

- For miRNA data: project, TCGA-LAML; data category, transcriptome profiling; data type, miRNA expression quantification; and experimental strategy, miRNA-Seq. Differentially expressed genes and miRNAs were selected as previously described.

## Network analysis for mRNA–miRNA interaction

TargetScan, an miRNA target gene database, was used in this study. TargetScan predicts the gene targets of miRNAs and provides raw data. A "Nonconserved_Site_Context_Scores.txt" file was downloaded from TargetScan, and the tab-delimited file was loaded as the "fread" function of the R package "data.table" (523.95 MB). From the "miRNA" column, the first three letters were used to identify species, and the remaining part was used as a key to identify miR-NAs. In this study, we used the miRNAs starting with "hsa," which refers to human miRNA. For the prediction of targets regulated by miRNAs, an miRNA–mRNA linked data frame was

used and merged with a dataframe containing genes and miRNAs. Two miRNA–mRNA pairs, our miRNA and TCGA mRNA, and TCGA miRNA and TCGA mRNA pairs were merged as a dataframe. The predicted target genes for each miRNA are listed in **Table 2**. Cytoscape (version 3.9.1) was used to visualize the identified genes and miRNAs in the interaction network. In the network, the background of genes and the font color of miRNAs are presented as a gradient according to the fold change.

### Visualization of miRNA expression levels and survival analysis using the TCGA database

TCGA-LAML miRNA expression data from 174 patients and their associated information, including risk stratification and survival, were used for visualization and survival analysis. Using the "beeswarm" function of the "beeswarm" R package, the miRNA expression levels of patients in the favorable and non-favorable groups were visualized as a dot plot. For survival analysis, the survival was presented using the "survfit" function of the "survival" R package. The median miRNA or mRNA expression levels were used to separate the participants into high-expression and low-expression groups. The pointed lines indicate the median survival rate. The log-rank test was performed, and the 95% confidence intervals are provided in the shade. The Kaplan–Meier plot was generated using the "ggsurvplot" function of the "survminer" R package. All parameters were used as default. The survival rates of each group are expressed using median and 95% confidence intervals.

## Results

### Validation of EVs derived from BM aspirates of the patients with AML

We used commercial SEC to isolate EVs and presented the separated EV profiles, as shown in **Fig 1,** using serum and plasma from a representative patient sample with this method. The average size of the BM aspirate-derived vesicles was 115.5 ± 2.7 nm in serum and 101.5 ± 4.4 nm in plasma; these sizes are within the size range of typical EVs (**Fig 1A and 1B**). The concentration information of EVs for each fraction is shown in **Fig 1A**. In TEM images, the size of the isolated vesicles was <200 nm, and they were visualized as cup-shaped vesicles under high magnification (**Fig 1C**). Western blotting revealed that the isolated vesicles were positive for EV markers (CD63 and CD81) in both serum and plasma (**Fig 1D** and **S1 Raw** images).

### Profiling miRNAs of EVs derived from BM aspirates of patients with AML

We compared the expression levels of EV-derived miRNAs between the favorable (n = 10) and non-favorable (intermediate and adverse, n = 22) risk groups. The total miRNA expression landscape according to risk groups is presented in **Fig 2**. A total of 965 EV-derived miRNAs were identified in all samples; 34 differentially expressed EV-derived miRNAs were identified in the non-favorable risk group (23 with downregulated expression and 11 with upregulated expression) (**S2** and **S3 Files**). Among these differentially expressed EV-derived miRNAs in the non-favorable risk group, the top ten miRNAs with significantly downregulated expression according to the level of fold change in the non-favorable groups, and all miRNAs with upregulated expression in the non-favorable group were evaluated as potential biomarker candidates for further analysis.

Additional analysis of differentially expressed EV-derived miRNA was conducted for patients who had achieved complete remission following induction chemotherapy or who had survived for more than one year following stem cell transplantation. The results of this analysis are presented in **S1** and **S2 Figs**, **S4** and **S5 Files.**

**Table 2. Differentially expressed miRNAs of EVs derived from BM aspirates of patients with AML.**

**(A) Downregulated miRNAs in the non-favorable risk group**

| miRNA name | Accession number | Predicted target genes | Fold change | P-value |
|---|---|---|---|---|
| hsa-miR-181b-5p | MIMAT0000257 | CALCRL, CD163, CRISPLD1, DDIT4, GPR126, HOXA11, HOXB5, INHBA, LGALSL, PLA2G4A, RAB27B, SCHIP1, SLC8A1 | -4.86595 | 0.002649 |
| hsa-miR-26b-5p | MIMAT0000083 | CALCRL, CARMIL1, DDIT4, GPR126, HOXA5, HOXA9, HOXB6, INHBA, KCNJ2, KCNK1, LRRC16, LRRC16A, MPEG1, RTN1 | -3.97035 | 0.011798 |
| hsa-miR-130a-3p | MIMAT0000425 | CLIP4, CRISPLD1, FAM155B, HOXA3, HOXA4, HOXA5, HTR1F, INHBA, KCNJ2, LGALSL, PDE7B, PRDM16, RTN1, SLC8A1, SORT1, SPOCK3, TMEM28, TSLP, ZFPM2 | -3.70651 | 0.019761 |
| hsa-miR-143-3p | MIMAT0000435 | ABLIM3, CD109, DHRS9, HOXA5, HTR7, IL2RA, INHBA, LRRC16A, PDE7B, SORT1, SRPX2, THBS1, TSLP, WDR49, ZFPM2 | -3.67306 | 0.017638 |
| hsa-miR-151b | MIMAT0010214 | (Not detected) | -3.28825 | 0.013929 |
| hsa-miR-3151-5p | MIMAT0015024 | (Not detected) | -3.24078 | 0.006773 |
| hsa-miR-224-5p | MIMAT0000281 | CPNE8, FHL1, HOXA11, HOXA5, HOXA7, IL1RN, KIAA0087, LGALSL, MEIS1, MPEG1, MS4A6A | -3.19268 | 0.007343 |
| hsa-miR-1255b-5p | MIMAT0005945 | (Not detected) | -3.16337 | 0.043645 |
| hsa-miR-151a-5p | MIMAT0004697 | (Not detected) | -3.13402 | 0.013409 |
| hsa-miR-26a-5p | MIMAT0000082 | CALCRL, CARMIL1, HOXA5, HOXA9, INHBA, KCNJ2, KCNK1, LRRC16, LRRC16A, RTN1 | -3.02336 | 0.038334 |
| hsa-miR-330-3p | MIMAT0000751 | ABHD17C, CLIP4, CPNE8, CRISPLD1, FAM108C1, GPR126, HOXA3, HOXA6, HOXA7, HTR7, INHBA, OLFML2A, PDE7B, PRDM16, SLC8A1, SORT1, SPOCK3, SYTL4, THBS1 | -2.93493 | 0.015633 |
| hsa-miR-4417* | MIMAT0018929 | (Not detected) | -2.89772 | 0.028909 |
| hsa-miR-106a-5p | MIMAT0000103 | ABLIM3, CD109, CLIP4, CNNM1, FHL1, HOPX, INHBA, KCNJ2, LAMA3, LGALSL, MMP7, MSR1, OLFML2A, PDE7B, PRDM16, RSNL2, RTN1, SPOCK3, SYTL4, THBS1, XIRP2, ZFPM2 | -2.74702 | 0.025757 |
| hsa-miR-25-5p | MIMAT0004498 | (Not detected) | -2.734 | 0.024185 |
| hsa-miR-130b-3p | MIMAT0000691 | AREG, CLIP4, CRISPLD1, FAM155B, HOXA3, HOXA4, HOXA5, HTR1F, INHBA, KCNJ2, LGALSL, LRRC16A, PDE7B, PRDM16, RTN1, SLC8A1, SORT1, SPOCK3, TMEM28, TSLP, ZFPM2 | -2.71124 | 0.007214 |
| hsa-miR-126-5p | MIMAT0000444 | CYP7B1, HOXA7 | -2.57377 | 0.023213 |
| hsa-miR-29c-5p | MIMAT0004673 | (Not detected) | -2.50307 | 0.046446 |
| hsa-miR-92a-1-5p | MIMAT0004507 | (Not detected) | -2.4919 | 0.045741 |
| hsa-miR-551a | MIMAT0003214 | (Not detected) | -2.35569 | 0.04152 |
| hsa-let-7d-5p | MIMAT0000065 | ABHD17C, HOXA10, HOXA5, HOXA9, HOXB6, ITGB3, S100A8, SAGE1, THBS1 | -2.31193 | 0.023155 |
| hsa-miR-4429 | MIMAT0018944 | FAM155B, HOXA10, ITGB3, KCNK1, MPV17L, SRPX2 | -1.96047 | 0.039084 |
| hsa-miR-320b | MIMAT0005792 | CTSL2, CTSV, FAM155B, HOXA10, HOXA3, ITGB3, KCNK1, MPV17L, MYOF, SCHIP1, SRPX2, SYTL4 | -1.88915 | 0.027898 |
| hsa-miR-548w | MIMAT0015060 | CALCRL, CD109, CPNE8, DDIT4, MSR1, RTN1 | -1.71528 | 0.040637 |

**(B) Upregulated miRNAs in the non-favorable risk group**

| miRNA name | Accession number | Predicted target genes | Fold change | P-value |
|---|---|---|---|---|
| hsa-miR-23b-5p | MIMAT0004587 | (Not detected) | 0.888698 | 0.042873 |
| hsa-miR-188-5p | MIMAT0000457 | GLIS3, STOX2 | 0.936382 | 0.044256 |
| hsa-miR-628-5p | MIMAT0004809 | LAMC3 | 0.982317 | 0.049352 |

*(Continued)*

**Table 2.** (Continued)

| | | | | |
|---|---|---|---|---|
| hsa-miR-3187-3p | MIMAT0015069 | (Not detected) | 1.149035 | 0.02537 |
| hsa-miR-501-3p | MIMAT0004774 | ADAMTS3, ARPP-21 | 1.178979 | 0.0261 |
| hsa-miR-577 | MIMAT0003242 | ADAMTS3, ARPP21, IGFBP5, SIX3, ZNF711 | 1.185345 | 0.023891 |
| hsa-miR-3916 | MIMAT0018190 | (Not detected) | 1.237243 | 0.022037 |
| hsa-miR-3686 | MIMAT0018114 | IRX5 | 1.455449 | 0.034108 |
| hsa-miR-100-5p | MIMAT0000098 | SLC14A1 | 1.749682 | 0.020838 |
| hsa-miR-432-5p | MIMAT0002814 | (Not detected) | 1.826905 | 0.011412 |
| hsa-miR-6788-5p | MIMAT0027476 | (Not detected) | 2.261496 | 0.002995 |

Note

* Dead miRNA from miRBase.

## Evaluation of selected miRNAs as biomarkers based on TCGA analysis

To evaluate the potential of the selected miRNAs as biomarkers, we investigated the correlation of these miRNAs with risk stratification and the survival of patients with AML using information from the TCGA database (**Table 3** and **S6 File**). Among the downregulated

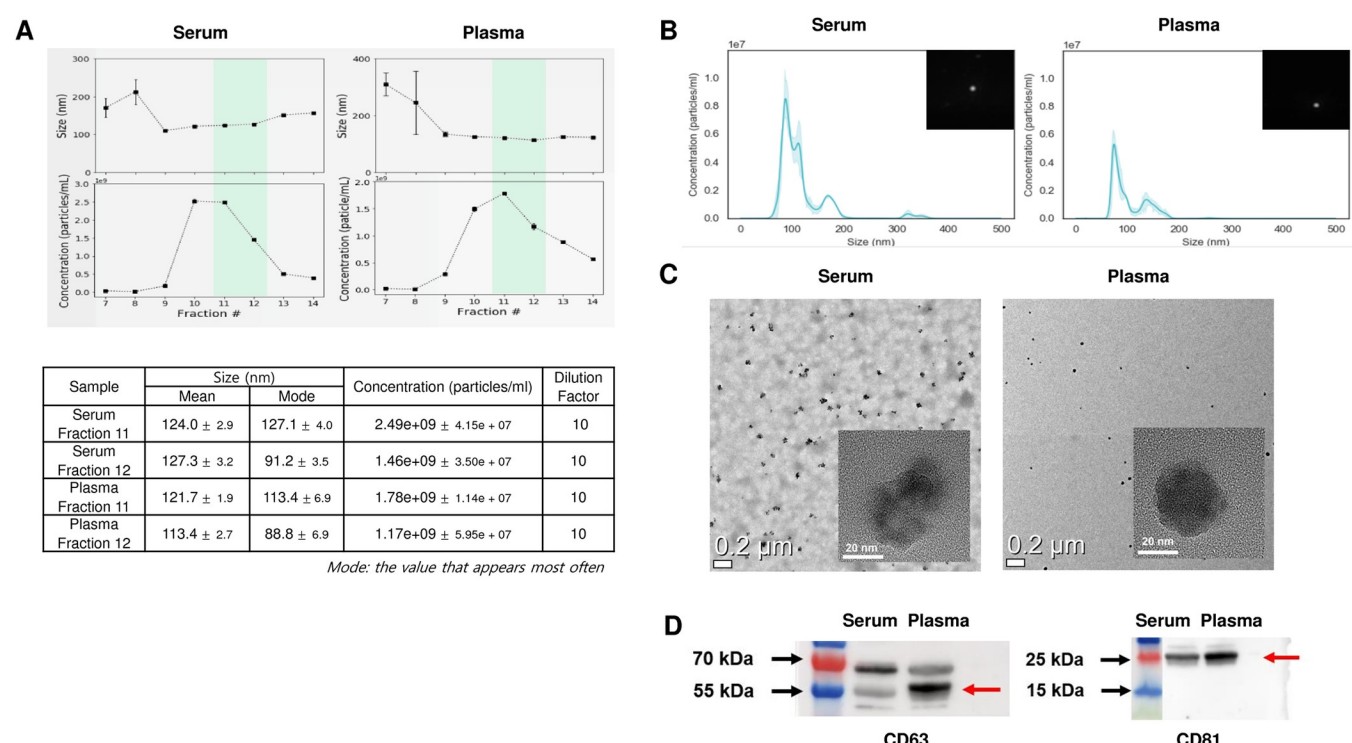

**Fig 1. Validation of extracellular Vesicles derived from BM aspirates of patients with AML.** (A) The size and concentration trend of vesicles by fractions are presented. The eluted fractions (11 and 12; 0.5 mL each) were used for vesicle isolation. Samples were diluted 10-fold. (B) The size distribution of the isolated vesicles was determined using nanoparticle tracking analysis (NTA). The average size of BM aspirate serum or plasma-derived vesicles was 115.5 ± 2.7 nm and 101.5 ± 4.4 nm, respectively; these sizes were within the size range of typical EVs. Samples were diluted 10-fold. (C) In transmission electron microscopy (TEM) images, the size of isolated vesicles was <200 nm, and they were visualized as cup-shaped vesicles under high magnification. (D) Western blotting showed that the isolated vesicles were positive for the markers of EVs (CD63 and CD81). EVs, extracellular vesicles; BM, bone marrow; AML, acute myelogenous leukemia.

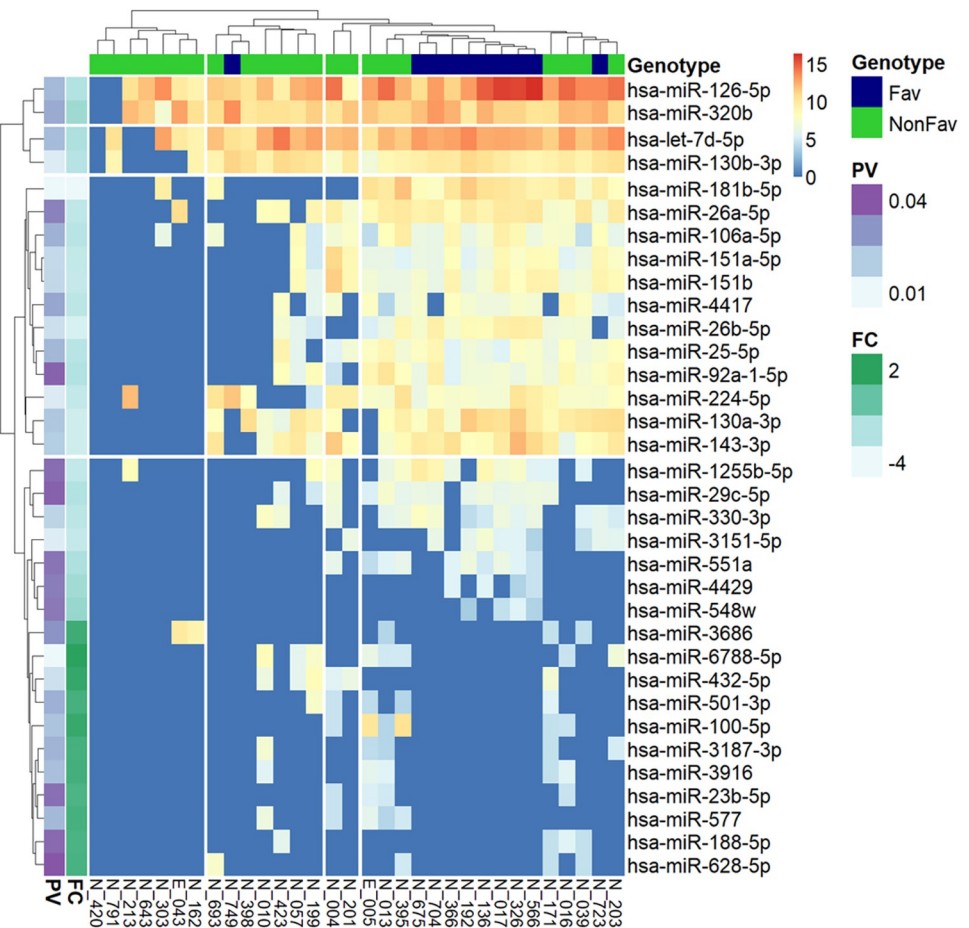

**Fig 2. Total miRNA expression landscape according to risk groups.** The expression levels of 34 differentially expressed miRNAs according to the favorable (n = 10) and non-favorable (intermediate and adverse, n = 22) risk groups based on the 2017 European LeukemiaNet recommendations are presented. The heatmap of 34 miRNAs (rows) from 32 patients (columns) shows the expression levels of each miRNA. Each column and row pair were clustered using the k-means clustering method with the package "pheatmap" in R and divided into four clusters. The column annotation bar indicates the favorable and non-favorable patients and the two-row annotation bars indicate the results from the *t*-test of p-value (PV) and fold change (FC) between the two genotypes (favorable and non-favorable). miRNA, microRNA.

miRNAs in the non-favorable risk group, hsa-miR-181b and hsa-miR-143 exhibited downregulated expression, and these correlated with non-favorable risk and short overall survival (**Fig 3A–3C**), whereas hsa-miR-130a and hsa-miR-224 showed downregulated expression and correlated with non-favorable risk, but not with overall survival. Regarding the miRNAs with upregulated expression in the non-favorable risk group, hsa-miR-188 and hsa-miR-501 demonstrated upregulated expression and correlated with non-favorable risk. There was no statistical significance in their correlation with overall survival when the dichotomy was conducted based on the survival period of 24 months (**Table 3**); however, their correlation with median survival rate according to follow-up time was statistically significant (**Fig 3D and 3E**).

## Network analysis of miRNAs–mRNAs for identifying candidate biomarkers

To extract more relevant mRNAs, miRNA–mRNA network analysis between EV-derived miRNAs in this study and mRNAs from the TCGA database was performed using Cytoscape

**Table 3. Evaluation of the selected miRNA as biomarker based on The Cancer Genome Atlas data.**

**(A) Downregulated miRNAs in the non-favorable risk group**

| | Risk stratification | | | Overall survival | | |
|---|---|---|---|---|---|---|
| | Favorable risk (median) | Non-favorable risk (median) | P-value | Overall survival ≥24 months (median) | Overall survival <24 months (median) | P-value |
| **hsa-mir-181b-1** | 11.8409 | 10.5012 | **<0.0001** | 11.5463 | 10.6491 | **0.0036** |
| **hsa-mir-181b-2** | 11.5922 | 9.936 | **<0.0001** | 10.9222 | 10.0078 | **0.0155** |
| hsa-mir-26b | 11.9759 | 11.9124 | 0.8589 | 11.8313 | 11.9662 | 0.1156 |
| **hsa-mir-130a** | 9.1187 | 7.8006 | **<0.0001** | 8.0383 | 8.1846 | 0.7682 |
| **hsa-mir-143** | 11.3432 | 10.0119 | **<0.0001** | 11.2577 | 10.1746 | **0.0002** |
| hsa-mir-151b | 0 | 1.2675 | 0.0132 | 0 | 1.2863 | 0.0335 |
| hsa-mir-3151 | 0 | 0 | 0.7761 | 0 | 0 | 0.2443 |
| **hsa-mir-224** | 3.8093 | 2.718 | **0.0002** | 3.5415 | 2.8095 | 0.0636 |
| hsa-miR-1255b-5p | — | — | — | — | — | — |
| hsa-mir-151a | 7.5796 | 8.2874 | 0.0326 | 7.9872 | 8.0932 | 0.2448 |
| hsa-mir-26a-1 | 11.2191 | 11.0118 | 0.1557 | 11.0423 | 11.0767 | 0.9281 |
| hsa-mir-26a-2 | 11.2088 | 10.9929 | 0.1276 | 10.9855 | 11.0617 | 0.9473 |

**(B) Upregulated miRNAs in the non-favorable risk group**

| | Risk stratification | | | Overall survival | | |
|---|---|---|---|---|---|---|
| | Favorable risk (median) | Non-favorable risk (median) | P-value | Overall survival ≥24 months (median) | Overall survival <24 months (median) | P-value |
| hsa-mir-23b | 8.4804 | 8.5984 | 0.8708 | 8.7242 | 8.5579 | 0.0314 |
| hsa-mir-188 | 1.642 | 2.9512 | 0.0013 | 2.1203 | 2.9297 | 0.1602 |
| hsa-mir-628 | 7.4201 | 7.6681 | 0.0121 | 7.6974 | 7.595 | 0.0401 |
| hsa-miR-3187-3p | 0 | 0 | 0.5977 | 0 | 0 | 0.1764 |
| hsa-mir-501 | 4.8439 | 5.9932 | 0.0001 | 5.0614 | 5.9115 | 0.2492 |
| hsa-mir-577 | 4.3508 | 3.1021 | 0.023 | 3.1002 | 3.383 | 0.9149 |
| hsa-mir-3916 | 0 | 0 | 0.1398 | 0 | 0 | 0.4567 |
| hsa-mir-3686 | 0 | 0 | 0.9312 | 0 | 0 | 0.4115 |
| hsa-mir-100 | 10.4656 | 8.1466 | <0.0001 | 9.9273 | 8.3651 | 0.0030 |
| hsa-mir-432 | 1.8731 | 0 | 0.0001 | 0 | 0 | 0.1250 |
| hsa-mir-6788 | 0 | 0 | <0.0001 | 0 | 0 | 0.2577 |

**Note:** Bold means statistical significance. Underbar refers to a case in which upregulation or downregulation was predicted in the non-favorable risk group based on the patient sample of this study; the opposite pattern was shown in the TCGA data analysis (Table 2).

(**Fig 4** and **Table 4**). Among the mRNAs showing interactions with biomarker candidate miR-NAs in this study (hsa-miR-181b, hsa-miR-143, hsa-miR-130a, hsa-miR-224, hsa-miR-188, and hsa-miR-501, **Table 2**), DDIT4, PLA2G4A, RAB27B, CD163, CALCRL, SLC8A1, CRISPLD1, SCHIP1, LGALSL, SORT1, PDE7B, HTR1F, CLIP4, PRDM16, RTN1, KCNJ2, CPNE8, KIAA0087, FHL1, STOX2, GLIS3, and ADAMTS3 were significantly correlated with the survival of patients with AML in the TCGA database analysis (**Fig 5**). The miRNA–mRNA network analysis between miRNAs and mRNAs from the TCGA database also showed a pattern similar to that observed in the findings from the miRNA–mRNA network analysis conducted between EV-derived miRNAs in this study and mRNAs from TCGA database (**S3 Fig**).

## Discussion

This study aimed not only to investigate the significance of EV-derived miRNAs as biomarkers but also to identify mRNAs with potential as biomarkers or druggable targets using EV-

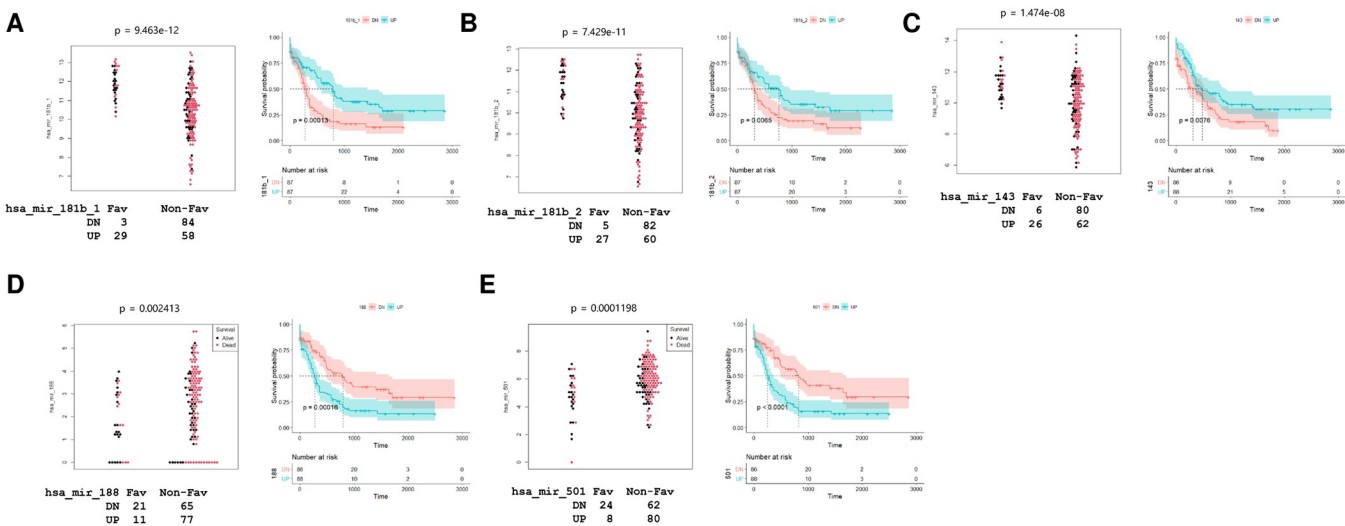

**Fig 3. Evaluation of selected miRNAs as biomarkers for the prediction of non-favorable risk and overall survival based on TCGA database.** Based on the results of TCGA analysis (Table 3), miRNAs of biomarker candidates for distinguishing between non-favorable risk and survival are presented. miRNA, microRNA; TCGA, The Cancer Genome Atlas.

derived miRNAs. We isolated EVs from BM aspirates of patients with AML using the high-purity method. We also investigated the potential of EV-derived miRNAs and that of mRNAs correlated with these EV-derived miRNAs as prognostic biomarkers for AML using the TCGA database. In this study, hsa-miR-181b, hsa-miR-143, hsa-miR-130a, and hsa-miR-224 were correlated with non-favorable risk. Of these, hsa-miR-181b and hsa-miR-143 were also

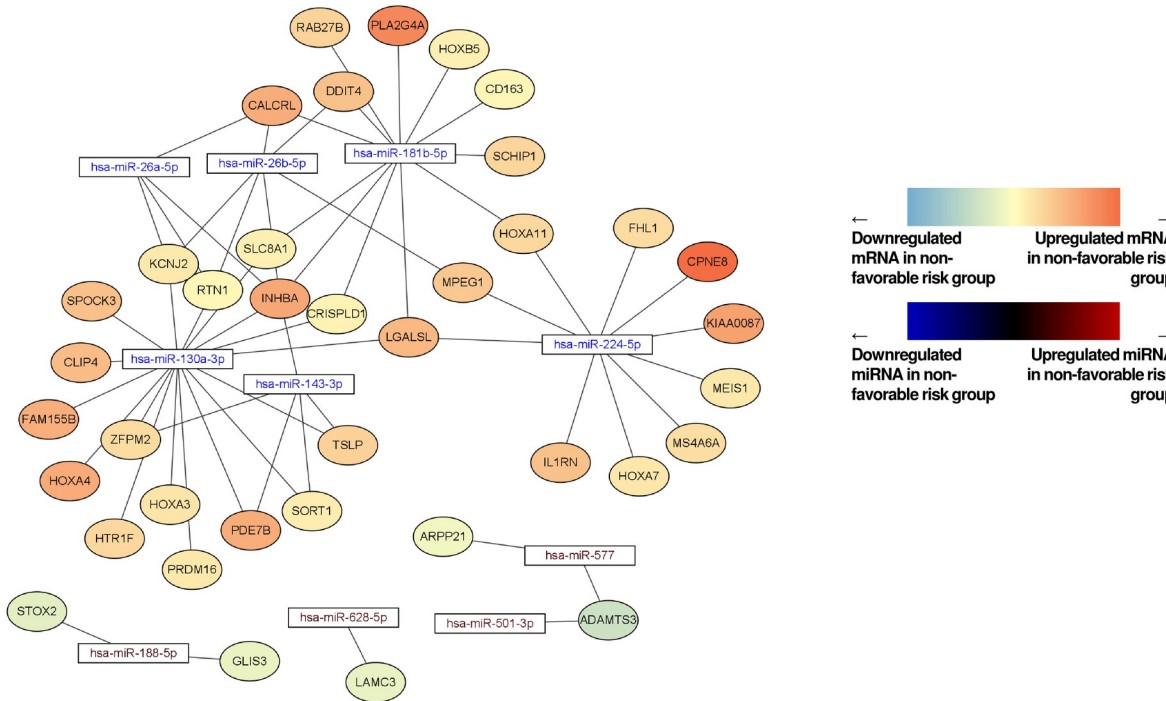

**Fig 4. Network analysis between EV-derived miRNA of this study and mRNA through the TCGA database analysis.** miRNA, microRNA; TCGA, The Cancer Genome Atlas.

**Table 4. miRNA–mRNA network analysis between EV-derived miRNA of this study and mRNA of the TCGA database.**

| miRNA | mRNA* | Mean (Favorable risk) | Mean (Non-favorable risk) | Fold change (Non-favorable risk/Favorable risk) | p-value |
|---|---|---|---|---|---|
| **hsa-mir-181b** | **DDIT4** | 1.974006396 | 3.498712495 | 1.524706099 | 9.08403E-08 |
| | **PLA2G4A** | 1.833803237 | 3.320603079 | 1.486799843 | 1.25147E-08 |
| | **RAB27B** | 1.349143863 | 2.503693526 | 1.154549663 | 4.66203E-06 |
| | **CD163** | 1.483401259 | 2.620549437 | 1.137148178 | 0.001698671 |
| | **CALCRL** | 0.611076075 | 1.629622854 | 1.018546779 | 2.30607E-08 |
| | **SLC8A1** | 0.199042712 | 0.9257533 | 0.726710588 | 9.50323E-13 |
| | **CRISPLD1** | 0.915036416 | 1.6345274 | 0.719490984 | 9.57523E-05 |
| | **SCHIP1** | 0.837771556 | 1.424835049 | 0.587063494 | 0.002934068 |
| | **LGALSL** | 0.377282508 | 0.868762463 | 0.491479954 | 3.92699E-07 |
| | HOXB5 | 0.091342269 | 1.634450721 | 1.543108452 | 2.89216E-18 |
| | HOXA11 | 0.06091293 | 0.720948304 | 0.660035374 | 2.26015E-07 |
| | INHBA | 0.215665581 | 0.624001139 | 0.408335558 | 1.30303E-05 |
| **hsa-mir-143** | **SORT1** | 1.865400582 | 2.707290427 | 0.841889845 | 0.015766943 |
| | **PDE7B** | 0.596505274 | 0.833058063 | 0.236552789 | 0.028201436 |
| | TSLP | 0.038498948 | 0.458173155 | 0.419674207 | 2.85219E-11 |
| | INHBA | 0.215665581 | 0.624001139 | 0.408335558 | 1.30303E-05 |
| | ZFPM2 | 0.046879955 | 0.217515321 | 0.170635366 | 0.001455456 |
| **hsa-mir-130a** | **HTR1F** | 0.484904255 | 1.886178298 | 1.401274043 | 7.41276E-09 |
| | **CLIP4** | 0.985565589 | 2.223365891 | 1.237800302 | 1.03933E-09 |
| | **PRDM16** | 0.022331028 | 1.167580436 | 1.145249407 | 4.3161E-15 |
| | **CRISPLD1** | 0.915036416 | 1.6345274 | 0.719490984 | 9.57523E-05 |
| | **LGALSL** | 0.377282508 | 0.868762463 | 0.491479954 | 3.92699E-07 |
| | **RTN1** | 0.146117872 | 0.599101859 | 0.452983987 | 1.71943E-06 |
| | **KCNJ2** | 0.084803523 | 0.41478121 | 0.329977687 | 3.20801E-08 |
| | **PDE7B** | 0.596505274 | 0.833058063 | 0.236552789 | 0.028201436 |
| | **SLC8A1** | 0.199042712 | 0.9257533 | 0.726710588 | 9.50323E-13 |
| | **SORT1** | 1.865400582 | 2.707290427 | 0.841889845 | 0.015766943 |
| | HOXA3 | 0.086023495 | 2.280483283 | 2.194459788 | 1.13279E-28 |
| | HOXA4 | 0.094582392 | 1.720972021 | 1.626389629 | 3.15222E-26 |
| | INHBA | 0.215665581 | 0.624001139 | 0.408335558 | 1.30303E-05 |
| | FAM155B | 0.05809143 | 0.2016298 | 0.14353837 | 0.022016379 |
| | SPOCK3 | 0.035294082 | 0.328054018 | 0.292759936 | 8.23759E-05 |
| | TSLP | 0.038498948 | 0.458173155 | 0.419674207 | 2.85219E-11 |
| | ZFPM2 | 0.046879955 | 0.217515321 | 0.170635366 | 0.001455456 |
| **hsa-mir-224** | **CPNE8** | 0.266893902 | 2.987551747 | 2.720657845 | 5.13835E-39 |
| | **LGALSL** | 0.377282508 | 0.868762463 | 0.491479954 | 3.92699E-07 |
| | **KIAA0087** | 0.013472336 | 0.42539671 | 0.411924374 | 3.13299E-05 |
| | **FHL1** | 2.157605344 | 2.445275081 | 0.287669738 | 0.207052121 |
| | MEIS1 | 0.998115171 | 2.735521646 | 1.737406475 | 1.34993E-06 |
| | HOXA7 | 0.042852188 | 1.575490649 | 1.532638461 | 2.75568E-26 |
| | MPEG1 | 3.74894259 | 5.078177875 | 1.329235285 | 0.030444314 |
| | MS4A6A | 3.00897467 | 3.818520257 | 0.809545587 | 0.08916414 |
| | HOXA11 | 0.06091293 | 0.720948304 | 0.660035374 | 2.26015E-07 |
| | IL1RN | 1.521855792 | 2.17705187 | 0.655196078 | 0.021926448 |
| **hsa-mir-188** | **STOX2** | 0.865028982 | 0.307821171 | -0.557207811 | 0.007630399 |
| | **GLIS3** | 0.571871548 | 0.141417862 | -0.430453686 | 0.000992895 |

*(Continued)*

**Table 4.** (Continued)

| miRNA | mRNA* | Mean (Favorable risk) | Mean (Non-favorable risk) | Fold change (Non-favorable risk/Favorable risk) | p-value |
|---|---|---|---|---|---|
| **hsa-mir-501** | **ADAMTS3** | 0.803756736 | 0.551538849 | -0.252217888 | 0.160393973 |

Note

* Bold means mRNA showing statistical significance in the survival of patients with AML through the TCGA analysis (Fig 5).

associated with poor overall survival. Upregulated expression of hsa-miR-188 and hsa-miR-501 was correlated with non-favorable risk, and these miRNAs were predictors of poor survival. In these EV-derived miRNA–mRNA network analyses using the TCGA database, we identified mRNAs with the potential to be used as biomarkers of poor prognosis.

Several studies have reported a correlation between downregulated miR-181b expression and poor prognosis in patients with AML [44, 45]. In a previous study, miR-181b expression was decreased in human multidrug-resistant leukemia cells and relapsed/refractory AML

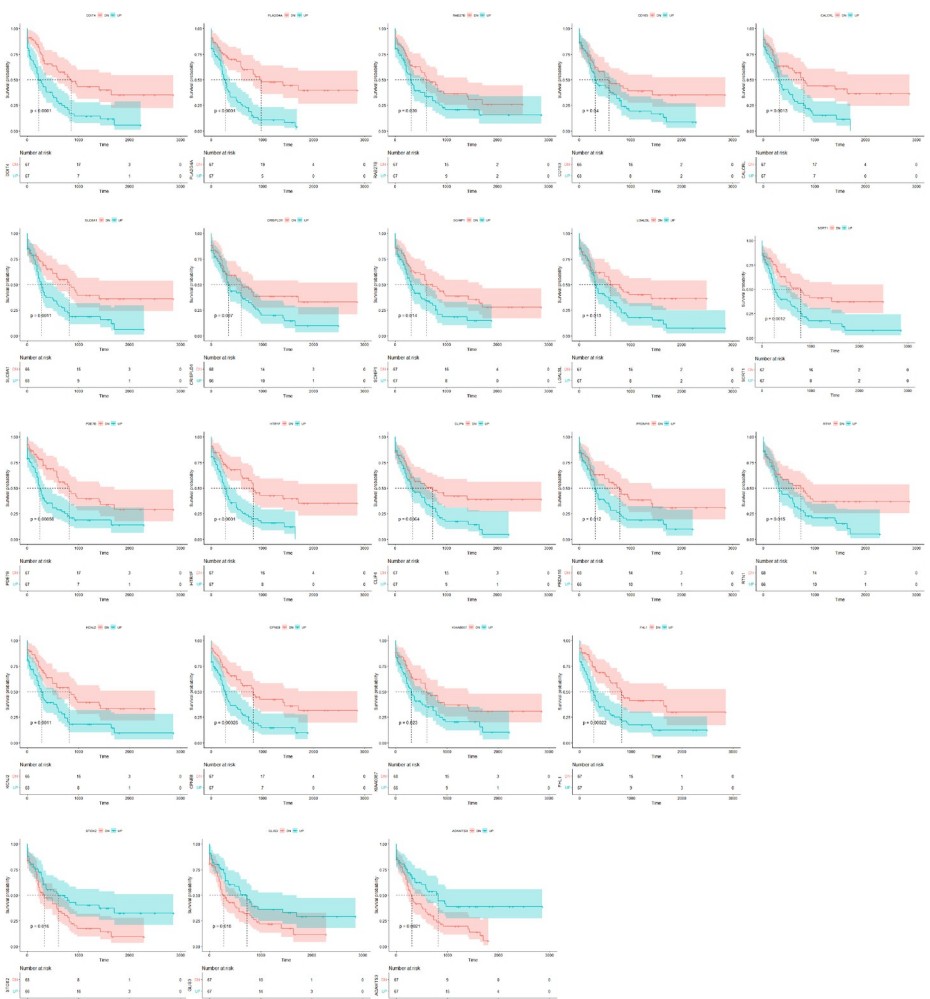

**Fig 5. Evaluation of candidate miRNA-associated mRNAs selected as biomarkers in this study for biomarkers or drug targets through TCGA database analysis.** miRNA, microRNA; TCGA, The Cancer Genome Atlas.

patient samples and overexpression of miR-181b was associated with increased drug sensitivity to cytotoxic chemotherapeutic agents and promoted drug-induced apoptosis [44]. A study of the BM aspirates of 84 patients with AML revealed a correlation between miR-181b insufficiency and poor risk stratification and that it was a predictor of treatment failure and poor survival [45]. Regarding miR-143, previous studies have reported its potential in the clinical detection of AML [46] and its association with poor survival when its expression is downregulated [47]. Although a few studies on AML have reported that miR-130a may act as a tumor suppressor gene or influence genes to enhance the sensitivity of drug-resistant cells in many cancers [48], some studies report that downregulation of miR-224 expression may have a role in cell survival and chemoresistance in chronic myeloid leukemia [49], and upregulated miR-188 expression may be associated with the survival of patients with AML [50]. Considering our study findings, an EV-derived miRNA-based analysis may yield findings similar to those obtained through cell-based miRNA assays.

Cancers involve complex ecosystems composed of tumor cells and a variety of non-cancerous cells, including various immune cell types, cancer-associated fibroblasts, endothelial cells, pericytes, and several other tissue-resident cell types [51]. Under these circumstances, there are various methods of intercellular communication between cancer cells and the tumor microenvironment, including EVs, which can serve as a target for liquid biopsy or therapeutic targets [52–54]. EVs can be released with the characteristics of the donor cells and can also arise in the context of specific microenvironmental dynamics [55, 56]. Ghetti et al. reported that circPVT1 is overexpressed by primary blast cells rather than in hematopoietic stem progenitor cells in patients with newly diagnosed AML. circPVT1 is released as cell-free RNA and in the form of EVs, suggesting a role for EVs in the crosstalk between AML cells and the microenvironment [57]. Furthermore, in a study comparing the EV miRNA profiles in the serum of AML patients (n = 5) and healthy volunteers (n = 5), specific miRNA patterns were observed in AML patients compared to those in healthy volunteers, indicating the potential utility of liquid biopsy [58]. EVs are known to be involved in AML stem cell maintenance [59], AML proliferation and progression [60], and even chemotherapy resistance [61]. EV-based analysis is also advantageous since it can be easily applied to study the dynamics of cancer in its microenvironment, as it requires a relatively minimal amount of information to analyze than does the cell-based approach. In addition, EV-encapsulated contents, including miRNAs, are highly protected from degradation and may be better biomarkers than non-EV-encapsulated ones [62]. Collectively, EVs may serve as an adjunct or potentially alternative means to a cell-based approach.

We performed an analysis based on TCGA data to identify mRNAs associated with miRNAs that were identified as potential biomarkers in this study. Twenty-one mRNA candidates used in this study are presented in **Fig 5**; some were reported to be biomarkers in previous studies (e.g., DDIT4 [63], PLA2G4A [64], RAB27B [65], CD163 [66], CALCRL [67], SLC8A1 [68], CRISPLD1 [69], PDE7B [70], PRDM16 [71], and KIAA0087 [72]), and some were reported to be novel target mRNAs or to require further validation for their role in AML (e.g., SCHIP1, LGALSL, SORT1, HTR1F, CLIP4, RTN1, KCNJ2, CPNE8, STOX2, GLIS3, and ADAMTS3). Almost all the above-mentioned studies were performed using cell-based mRNA analyses. In this regard, our study is considered meaningful because its findings suggest that we could obtain results from EV-derived miRNA-based and bioinformatics analyses similar to those obtained from cell-based miRNA and mRNA analyses. Considering the characteristics of EVs, which summarize the core characteristics of donor cells, we believe that this EV-derived miRNA-based analysis approach may be more efficient than the cell-based method as it can extract meaningful genes more compactly than the cell-based method. In addition,

biomarkers detected using this method may be potential drug targets; therefore, we believe that our findings will be of great help in terms of the efficiency of new drug development.

This study had several limitations. First, the study had a small sample size. To compensate for this, we used information from the TCGA database to validate the selected biomarkers in this study. Nevertheless, EV-derived miRNA-specific biomarkers with a different value from that obtained in a cell-based analysis may have been missed because the TCGA database offers cell-based miRNA values. Second, it should also be noted that the interpretation of the results is limited by the EV separation method. To date, there is no singular best method for EV isolation and each study should report the source of EV-containing materials and all methodological details of sample collection and EV isolation, and interpret the results under these conditions [73]. Although the SEC method was not included, a study comparing different EV isolation methods using bone marrow samples from AML patients showed that the method of EV isolation significantly impacts the yield and potential functionality of leukemia-derived EVs [74]. Third, there are limitations in the interpretation of the results owing to the complexity of the mechanism of action of miRNAs. The central role of miRNAs is to regulate gene expression through canonical and non-canonical mechanisms [75]. However, more than half of these processes are performed through non-canonical mechanisms, meaning that they are not always complementary [75]. Each miRNA can act on several mRNAs. In contrast, various mRNAs can interfere with a single miRNA. Therefore, further research is needed to determine whether the properties of EV-derived miRNA-based identification of miRNA or mRNA biomarkers presented in this study are homeostatic or variable according to the microenvironment. Nevertheless, as mentioned above, this study is meaningful in that it showed that EV-derived miRNA-based and bioinformatics analyses of related mRNAs may draw conclusions similar to those drawn using cell-based miRNA or mRNA analyses. EV-derived miRNA-based analysis can be easily applied to new research fields or clinical practice because the amount of information to be analyzed is relatively smaller than that obtained using the cell-based approach.

## Conclusions

EVs can be released with the characteristics of the AML cells and can also be involved in the context of specific microenvironmental dynamics between AML cells and the microenvironment. In this study, we isolated EVs from the BM of patients with AML using the SEC method and analyzed EV-derived miRNAs to determine their potential as a novel biomarker discovery method. Overall, the findings of this study revealed that EV-derived miRNAs (hsa-miR-181b, hsa-miR-143, hsa-miR-130a, hsa-miR-224, hsa-miR-188, and hsa-miR-501) may be biomarkers for risk stratification and prognostic prediction in AML. In addition, this study showed that EV-derived miRNA-based analysis may lead to conclusions similar to those drawn using cell-based miRNA or mRNA analyses. Furthermore, because it produces a relatively small amount of data, it can be easily applied to new research areas or clinical practice. However, further studies are required to validate these EV-derived miRNA-based analysis results in other large cohorts before they can be implemented in clinical practice.

## Supporting information

**S1 Fig. Total miRNA expression landscape based on the achievement of complete remission following induction chemotherapy.** G1: Patients who achieve a complete remission following induction chemotherapy. G2: Patients who fail to achieve a complete remission following induction chemotherapy. PV, P-value; FC, Fold change.
(TIF)

**S2 Fig. Total miRNA expression landscape according to patients with relapse-free survival >1 year after stem cell transplantation.** G1: Patients with relapse-free survival >1 year after stem cell transplantation. G2: Patients with relapse-free survival <1 year after stem cell transplantation. PV, P-value; FC, Fold change. Note: This analysis was performed on 17 patients who underwent stem cell transplantation among all patients.
(TIF)

**S3 Fig. Network analysis between miRNA and mRNA using TCGA database.** miRNA, microRNA; TCGA, The Cancer Genome Atlas.
(TIF)

**S1 Raw images.**
(PDF)

**S1 File.**
(XLSX)

**S2 File.**
(PDF)

**S3 File.**
(XLSX)

**S4 File.**
(XLSX)

**S5 File.**
(XLSX)

## Author Contributions

**Conceptualization:** Ka-Won Kang, Sunghoi Hong, Hyun Koo Kim, Yeonho Choi, Ji-ho Park, Yong Park.

**Data curation:** Ka-Won Kang.

**Formal analysis:** Ka-Won Kang, Jeong-An Gim, Ji-ho Park.

**Funding acquisition:** Yeonho Choi, Yong Park.

**Investigation:** Ka-Won Kang, Jeong-An Gim, Sunghoi Hong, Hyun Koo Kim, Yeonho Choi, Ji-ho Park.

**Methodology:** Ka-Won Kang, Jeong-An Gim, Ji-ho Park.

**Project administration:** Sunghoi Hong, Yeonho Choi, Yong Park.

**Resources:** Yong Park.

**Supervision:** Sunghoi Hong, Hyun Koo Kim, Yeonho Choi, Ji-ho Park, Yong Park.

**Visualization:** Ka-Won Kang.

**Writing – original draft:** Ka-Won Kang, Jeong-An Gim.

**Writing – review & editing:** Ka-Won Kang, Jeong-An Gim.

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
