## [Decision Letter · Decision Letter 0]

15 Feb 2024

PONE-D-23-13124

Use of extracellular vesicle microRNA profiles in patients with acute myeloid leukemia for the identification of novel biomarkers

PLOS ONE

Dear Dr. Park,

Thank you for submitting your manuscript to PLOS ONE. After careful consideration, we feel that it has merit but does not fully meet PLOS ONE’s publication criteria as it currently stands. Therefore, we invite you to submit a revised version of the manuscript that addresses the points raised during the review process.

Your manuscript provides an interesting proof of concept regarding microRNA isolation and evaluation from extracellular vesicles (EV) derived from Bone marrow samples in 32 Korean patients with newly diagnosed Acute Myeloid Leukemia (AML), specifically regarding its prognostic and predictive role in a ten-year timeframe. Besides incorporating a high purity isolation techniques, it presents intriguing findings with potential clinical significance for risk stratification and prognosis in AML. We highlight that the Bioinformatics analysis methods employed in the study have revealed meaningful potential regulatory relationships. Additionally, gel chromatography for EV isolation and survival analyses based on TCGA-LAML data are notable methodological approaches. We do believe these suggestions could improve the manuscript's quality and readiness for publication. Please consider the following main points to address:

1. Lack of clarity and detail regarding sample collection, processing, and preservation, which could impact the study's validity, especially considering the potential difficulty in identifying exosomes in samples preserved for an extended period. Please do provide comprehensive information regarding risk assessment and prognosis in the study population so to consider the translational relevance of your data.

2. The need for further justification of methodological choices, such as gel chromatography for EV isolation, and exploration of additional methods for identifying target genes of key miRNAs.

3. Incomplete reporting of differentially expressed miRNAs and suggestions for better presentation of data, such as through supplemental material or figures.

4. Requests for additional analyses on screened key miRNAs and a more comprehensive discussion, including the significance of the study's findings in the context of AML cell exosome functions and implications for the bone marrow microenvironment and tumor resistance.

We look forward to receiving your revised manuscript.

Kind regards,

Andres Mauricio Acevedo-Melo, M.D.

Academic Editor

PLOS ONE

Journal Requirements:

7. PLOS ONE now requires that authors provide the original uncropped and unadjusted images underlying all blot or gel results reported in a submission’s figures or Supporting Information files. This policy and the journal’s other requirements for blot/gel reporting and figure preparation are described in detail at https://journals.plos.org/plosone/s/figures#loc-blot-and-gel-reporting-requirements and https://journals.plos.org/plosone/s/figures#loc-preparing-figures-from-image-files. When you submit your revised manuscript, please ensure that your figures adhere fully to these guidelines and provide the original underlying images for all blot or gel data reported in your submission. See the following link for instructions on providing the original image data: https://journals.plos.org/plosone/s/figures#loc-original-images-for-blots-and-gels. 

Reviewers' comments:

Reviewer's Responses to Questions

**Comments to the Author**

1. Is the manuscript technically sound, and do the data support the conclusions?

Reviewer #1: Partly

Reviewer #2: Partly

Reviewer #3: Partly

2. Has the statistical analysis been performed appropriately and rigorously? 

Reviewer #1: Yes

Reviewer #2: Yes

Reviewer #3: Yes

3. Have the authors made all data underlying the findings in their manuscript fully available?

Reviewer #1: Yes

Reviewer #2: Yes

Reviewer #3: Yes

4. Is the manuscript presented in an intelligible fashion and written in standard English?

Reviewer #1: Yes

Reviewer #2: Yes

Reviewer #3: Yes

5. Review Comments to the Author

Reviewer #1: The Authors presented interesting results about differentially expressed vesicular miRNA that may serve as risk markers in adult AML patients. The Authors focused on EVs circulating in the BM and performed the analysis by NGS.

I have some comments:

- AML are conventionally considered a group of disorders, so i suggest the Authors to refer to AML as plural.

- How were the samples stored? Was the BM stored post centrifugation (meaning BM plasma) or as whole blood?

- Page 12. The Authors reported 2 average sizes of BM derived vesicles, but only 1 cohort is present. What do they mean? What does "respectively" refer to?

- Why did not the Authors consider also the miRNA differentially expressed in the favourable group as positive markers?

- I strongly suggest to analyze the correlation between the differentially expressed miRNA and the type of chemotherapy. Moreover, I am curious to investigate when the success of allogeneic transplant correlates with some altered expression of vesicular miRNA.

- In the discussion section, please add some comments concerning the application of the presented findings in translational medicine for liquid biopsy purpose. Some other Groups suggested to search leukemic markers in EVs of patients with AML, such as PMID 36263223, PMID 37536029, PMID 36926576, and PMID 36139669. Please, add a comment on these papers.

- Finally I suggest the Authors to stress why are they searching miRNA within the EVs released by cells that are present in the same samples (I mean, BM samples).

I hope my comments and suggestion will help the Authors in improving the quality of the manuscript.

Reviewer #2: In this study, the authors establish clinically significant microRNA sets using EVs from BM aspirates of patients with AML, and to identify the genes that interact with these EVs derived miRNAs in AML. They findings of this study revealed that EV-derived miRNAs (hsa-mir-181b, hsamir-143, hsa-mir-130a, hsa-mir-224, hsa-mir-188, and hsa-mir-501) may be biomarkers for risk stratification and prognosis in AML. In summary, there are several issues that require attention before the manuscript is considered ready for publication. The following is a list, not prioritized by importance, of these concerns.

- Introduction section: …accumulation of somatically acquiredgenetic alterations… include a reference

- Introduction section: …and may be useful biomarkers for the early detection of AML recurrence or prognosis… include a reference

- Introduction section:….used EV isolation methods, such as sequential ultracentrifugation or precipitation methods, with a relatively low purity….include a reference

- Review the abbreviations throughout the document: EV or EVs

- The authors could provide more information regarding the methods employed to determine the risk stratification and overall survival of the samples.

- Specify the method for quantifying the bands in the Western Blot

- Results Section, Table 1: Abbreviations such as EVs, BM... are not found within the table

- Discussion section: …between downregulated miR-181b expression and poor prognosis in patients with AML…… include a reference.

- Discussion section: …The central role of miRNAs is to regulate gene expression through canonical and non-canonical mechanisms…… include a reference.

- Discussion section: The sentence: “This study aimed to report not only the significance of EV-derived miRNA as a biomarker, but also to identify mRNAs with potential as biomarkers”…should go at the beginning of the discussion section

- Discussion section: …Several studies have reported a correlation between downregulated miR-181b expression and poor prognosis in patients with AML… include a reference.

- The quality of presented EVs images needs enhancement due to suboptimal resolution and sharpness.

- For the WB analysis, it is imperative to incorporate a positive control to validate the protein bands' association with the isolated EVs.

- Include the complete membrane image as a supplementary figure in the WB analysis, not just a fraction of the identified protein.

- In the WB analysis, include the MW bands to confirm that the identified protein aligns with the expected molecular weight.

- The authors are encouraged to incorporate a functional experiment involving a cell line, including an inhibitor for miR-188, miR-501, or a miR-181b mimic.

- Additionally, a functional study or gene expression analysis should be included in the analyzed samples to validate the identified target genes.

- Regarding the statement in the discussion, 'Each miRNA can act on several mRNAs. In contrast, various mRNAs can interfere with a single miRNA,' it is recommended that the authors provide further clarification or expand on this concept, as it may be perceived as confusing in its current form.

Reviewer #3: This manuscript's considerable interest lies in identifying essential miRNAs by obtaining EVs from bone marrow samples of AML patients, and bioinformatics analysis methods have brought several meaningful potential regulatory relationships. However, it is necessary to mention that there is still more excellent value in this study that deserves the authors' reflection and additions, and some critical problems still need to be resolved.

1. This study collected biological samples from 32 AML patients from 2008-2017, but information on sample collection, processing, and preservation is missing, and of particular concern is the difficulty of exosome identification if the samples have been preserved for an extended period, or in samples that have been preserved for 10 years. In addition, no clear instructions are given on the volume of samples to be collected per sample.

2. During the isolation of EVs, the authors used gel chromatography, which was a critical method in this study. The advantage of using SEC isolation is that the polymerization products of proteins and EVs are minimal. The functionality of the EVs is usually unaffected. However, it isn't easy to differentiate between particles of similar size but different characteristics by filtration or SEC separation methods based on component size. Thus, the authors should describe the reasons for choosing the SEC method concerning the purpose and needs of the study.

3. The target mRNAs of miRNAs have been established in several databases by experimental validation or computer prediction, and the data on the target genes of miRNAs available in different databases are not uniform, mainly in prediction methods, quantity, type, and elasticity of recorded data. Regarding the prediction of miRNA target genes, the authors used the TargetScan database, which is known to contain data based on the sequencing technique of 3P-seq to identify the corresponding 3'UTR region of a transcript and then to predict the target genes through the thermodynamic modeling of the interactions between the RNAs in conjunction with the analysis of the sequence comparisons. However, the authors can still try to apply more ways to explore the target genes of key miRNAs, for example, the multiMiR package, to obtain more prediction results and then carry out work such as miRNA bio-function or drug target analysis based on interaction analysis results.

4. The authors performed survival analyses based on clinical data from TCGA-LAML, suggesting that the list reports basic clinical information for the 174 AML patients used in this study.

5. The authors extracted exosomes and performed exosome particle size and marker assays in the study. Still, since SEC is a separation methodology approach based on size and the authors used two layers of column beads consisting of CL-6B sepharose and sephacryl s-200 high resolution to try to avoid the possibility of other contaminants of similar sizes that were present in the separation process, then the authors should have reported the results of the exosome concentration data.

6. By differential expression analysis of EVs-derived miRNAs, the authors obtained 965 EVs-derived miRNAs, but the authors listed only 32 DEmiRNAs in the non-favorable risk group, which is incomplete, and it is recommended that the authors provide the data results of all the differentially expressed miRNAs or the expression matrix of the sequencing results of the miRNAs in the form of supplemental material. Similarly, the up-and down-regulated significant miRNAs could be presented as figures based on the ggplot2 or heatmap package.

7. Presenting the results of survival analyses in table form is encouraged, but attention needs to be paid to the form and readability of the table, and a formal revision of Table 3 and Table 4 is needed.

8. The authors could have carried out more analyses on the screened key miRNAs, and the research methods provided in the current study may be too simplistic in the bioinformatics section. It is recommended to add, for example, analyses of GO, KEGG, Reactome, drug prediction, immune infiltration, and other multiple biological regulatory functions for miRNAs and the target genes or to increase the level of the analysis by further constructing the clinical prediction model based on the miRNAs to emphasize the biological significance of the key miRNAs.

9. Regarding the discussion section of the study, the authors are required to summarise and analyze the important results found to highlight the significance of this study. As it is well known that the exosomes of AML cells have aberrant biogenesis, secretion, and uptake, also the functions are of great interest, especially in the context of the exosomes of AML stem cells in creating an adaptive bone marrow microenvironment, tumor resistance, and extramedullary infiltration, the authors still need to address the contributions of the study and future challenges in the context of their important findings.

6. PLOS authors have the option to publish the peer review history of their article (what does this mean?). If published, this will include your full peer review and any attached files.

Reviewer #1: No

Reviewer #2: No

Reviewer #3: **Yes: **Jie Wang

---

## [Author Response · Author response to Decision Letter 0]

4 Apr 2024

Manuscript ID: PONE-D-23-13124

Title: Use of extracellular vesicle microRNA profiles in patients with acute myeloid leukemia for the identification of novel biomarkers

Dear Prof. George Vousden:

Thank you for your help in processing our manuscript, and we appreciate the reviewers’ thoughtful comments. First, we have formatted the manuscript according to the style requirements of PLOS ONE and modified content regarding the Data Availability statement and ethics statement. Unadjusted blot images were uploaded in Supporting Information files. We have also revised the manuscript in accordance with the reviewers’ comments and suggestions. Point-by-point responses are included in this letter to each comment of the reviewers. Our revisions address all points raised by the reviewers. Additionally, our manuscript has been reviewed by an English-speaking editor.

We believe that changes made in response to the reviewers’ comments have significantly improved the revised manuscript, and we hope that the revised manuscript is now suitable for publication.

Sincerely yours,

Yong Park, MD, Ph.D. 

Department of Internal Medicine, Korea University College of Medicine 73 Goryeodae-ro, Seongbuk-gu, Seoul 02841, Republic of Korea

Tel: +82-2-920-6847, Fax: +82-2-920-6520, E-mail: paark76@hanmail.net

---

## [Decision Letter · Decision Letter 1]

30 Apr 2024

PONE-D-23-13124R1Use of extracellular vesicle microRNA profiles in patients with acute myeloid leukemia for the identification of novel biomarkersPLOS ONE

Dear Dr. Park,

Thank you for submitting your manuscript to PLOS ONE. After careful consideration, we feel that it has merit but does not fully meet PLOS ONE’s publication criteria as it currently stands. Therefore, we invite you to submit a revised version of the manuscript that addresses the points raised during the review process.

Dear Authors:

Please provide answer to requested analysis and its impact as per study main objective. Please also ensure that the submitted information is readible as tables or figures (S2_AML_RPM_Genotype). Thanks

We look forward to receiving your revised manuscript.

Kind regards,

Andres Mauricio Acevedo-Melo, M.D.

Academic Editor

PLOS ONE

Journal Requirements:

Reviewers' comments:

Reviewer's Responses to Questions

**Comments to the Author**

1. If the authors have adequately addressed your comments raised in a previous round of review and you feel that this manuscript is now acceptable for publication, you may indicate that here to bypass the “Comments to the Author” section, enter your conflict of interest statement in the “Confidential to Editor” section, and submit your "Accept" recommendation.

Reviewer #1: (No Response)

Reviewer #3: All comments have been addressed

2. Is the manuscript technically sound, and do the data support the conclusions?

Reviewer #1: Yes

Reviewer #3: Yes

3. Has the statistical analysis been performed appropriately and rigorously? 

Reviewer #1: Yes

Reviewer #3: Yes

4. Have the authors made all data underlying the findings in their manuscript fully available?

Reviewer #1: Yes

Reviewer #3: Yes

5. Is the manuscript presented in an intelligible fashion and written in standard English?

Reviewer #1: Yes

Reviewer #3: Yes

6. Review Comments to the Author

Reviewer #1: The Authors partially reply to my concerns.

In particular, no analysis in the favorable group was made. The reply of the Authors is not sufficient. The favorable group could be considered in order to define a "protective" setting of miRNA.

Moreover, despite the consideration that the samples were collected at diagnosis, the impact of the information must not be limited at that time. miRNA are considered as predictive of different variables, included the response to therapy. A correlation between the basal level of miRNA expression and the response to therapy/HSCT is required.

Reviewer #3: The authors seem to have done their best to address my comments, and in subsequent studies I hope that the authors will carry out the appropriate biofunctional analyses for the key miRNAs they have found.

7. PLOS authors have the option to publish the peer review history of their article (what does this mean?). If published, this will include your full peer review and any attached files.

Reviewer #1: No

Reviewer #3: **Yes: **Jie Wang

---

## [Author Response · Author response to Decision Letter 1]

4 Jun 2024

Manuscript ID: PONE-D-23-13124R1

Title: Use of extracellular vesicle microRNA profiles in patients with acute myeloid leukemia for the identification of novel biomarkers

Dear Prof. George Vousden:

We appreciate your assistance in processing our manuscript and the insightful comments and suggestions provided by the editor and reviewers. We have incorporated the suggested modifications and included them below. Additionally, our manuscript has been reviewed by an English-speaking editor. Thank you for your guidance, and we hope to hear from you soon regarding the acceptance of our manuscript for publication.

Sincerely yours,

Yong Park, MD, Ph.D. 

Department of Internal Medicine, Korea University College of Medicine 73 Goryeodae-ro, Seongbuk-gu, Seoul 02841, Republic of Korea

Tel: +82-2-920-6847, Fax: +82-2-920-6520, E-mail: paark76@hanmail.net

 

List of reviewers’ comments and point-by-point responses

In each response, page and line numbers refer to the revised manuscript.

In the revised manuscript, text changes are highlighted in yellow. 

Editor:

1. Please provide answer to requested analysis and its impact as per study main objective. 

Response: According to the reviewer, we have included the analysis and described the relevant content in the manuscript (Page 15, lines 15–19).

2. Please also ensure that the submitted information is readible as tables or figures (S2_AML_RPM_Genotype).

Response: Sorry for the confusion. We have converted the format of the file to an Excel file so that you can review the data in each column (S2_AML_RPM_Genotype).

Reviewer #1:

1. The Authors partially reply to my concerns. In particular, no analysis in the favorable group was made. The reply of the Authors is not sufficient. The favorable group could be considered in order to define a "protective" setting of miRNA. Moreover, despite the consideration that the samples were collected at diagnosis, the impact of the information must not be limited at that time. miRNA are considered as predictive of different variables, included the response to therapy. A correlation between the basal level of miRNA expression and the response to therapy/HSCT is required.

Response: We would like to express our gratitude for your valuable feedback. Additional analysis of differentially expressed EV-derived miRNA was conducted for patients who had achieved complete remission following induction chemotherapy or who had survived for more than one year following stem cell transplantation (Supplementary figure 1 and 2, S4_AML_RPM_CR_after_induction_therapy, and S5_AML_RPM_Transplantation in Supporting Information files). Unfortunately, the number of bone marrow samples from patients who underwent transplantation was too small to yield significant results. Although the miRNA expression landscape based on the achievement of complete remission following induction chemotherapy showed relatively distinct patterns, we were unable to validate these findings due to the lack of corresponding information in the TCGA dataset. We have included the additional analysis and related files in the manuscript and presented them as supplementary files (Page 15, lines 15–19). 

Reviewer #3:

1. The authors seem to have done their best to address my comments, and in subsequent studies I hope that the authors will carry out the appropriate biofunctional analyses for the key miRNAs they have found.

Response: We are grateful for your valuable input. Moving forward, we will be committed to conducting the necessary biofunctional analyses for the key miRNAs identified in subsequent studies.

---

## [Decision Letter · Decision Letter 2]

26 Jun 2024

Use of extracellular vesicle microRNA profiles in patients with acute myeloid leukemia for the identification of novel biomarkers

PONE-D-23-13124R2

Dear Dr. Park,

We’re pleased to inform you that your manuscript has been judged scientifically suitable for publication and will be formally accepted for publication once it meets all outstanding technical requirements.

Kind regards,

Andres Mauricio Acevedo-Melo, M.D.

Academic Editor

PLOS ONE

Additional Editor Comments (optional):

Dear Authors.

Thank you for providing a revised version of your manuscript. Please provide commitment to perform a functional study or gene expression analysis in a follow-up study among risk subgroups as you have stated in response to reviewers' comments.

Best regards

Andrés M. Acevedo MD MSc

Reviewers' comments:

Reviewer's Responses to Questions

**Comments to the Author**

1. If the authors have adequately addressed your comments raised in a previous round of review and you feel that this manuscript is now acceptable for publication, you may indicate that here to bypass the “Comments to the Author” section, enter your conflict of interest statement in the “Confidential to Editor” section, and submit your "Accept" recommendation.

Reviewer #1: All comments have been addressed

Reviewer #2: All comments have been addressed

Reviewer #3: All comments have been addressed

2. Is the manuscript technically sound, and do the data support the conclusions?

Reviewer #1: Yes

Reviewer #2: Partly

Reviewer #3: Yes

3. Has the statistical analysis been performed appropriately and rigorously? 

Reviewer #1: Yes

Reviewer #2: Yes

Reviewer #3: Yes

4. Have the authors made all data underlying the findings in their manuscript fully available?

Reviewer #1: Yes

Reviewer #2: Yes

Reviewer #3: Yes

5. Is the manuscript presented in an intelligible fashion and written in standard English?

Reviewer #1: Yes

Reviewer #2: Yes

Reviewer #3: Yes

6. Review Comments to the Author

Reviewer #1: (No Response)

Reviewer #2: The authors have partially addressed my comments, but no additional studies were conducted on the favorable group. I hope they will perform the appropriate biofunctional analysis for the key miRNAs they have identified.

Reviewer #3: (No Response)

7. PLOS authors have the option to publish the peer review history of their article (what does this mean?). If published, this will include your full peer review and any attached files.

Reviewer #1: **Yes: **Simona Bernardi

Reviewer #2: No

Reviewer #3: No

---

## [Editor Report · Acceptance letter]

13 Aug 2024

PONE-D-23-13124R2 

PLOS ONE

Dear Dr. Park, 

I'm pleased to inform you that your manuscript has been deemed suitable for publication in PLOS ONE. Congratulations! Your manuscript is now being handed over to our production team.

Kind regards, 

on behalf of

Dr. Andres Mauricio Acevedo-Melo 

Academic Editor

PLOS ONE